



# A Method for Computing the Three-Dimensional Radial Distribution Function of Cloud Particles from Holographic Images

Michael L. Larsen[1,2] and Raymond A. Shaw[2]

[1]Department of Physics and Astronomy, College of Charleston, Charleston, SC, USA
[2]Department of Physics, Michigan Technological University, Houghton, MI, USA

**Correspondence:** Michael L. Larsen (LarsenML@cofc.edu)

**Abstract.** Reliable measurements of the three-dimensional radial distribution function for cloud droplets are desired to help characterize microphysical processes that depend on local drop environment. Existing numerical techniques to estimate this three-dimensional radial distribution function are not well suited to *in situ* or laboratory data gathered from a finite experimental domain. This manuscript introduces and tests a new method designed to reliably estimate the three-dimensional radial distribution function in contexts where (i) physical considerations prohibit the use of periodic boundary conditions and (ii) particle positions are measured inside a convex volume that may have a large aspect ratio. The method is then utilized to measure the three-dimensional radial distribution function from laboratory data taken in a cloud chamber from the Holographic Detector for Clouds (HOLODEC).

## 1 Introduction

Cloud droplet clustering is relevant to physical processes like condensational growth (e.g. Srivastava (1989); Kostinski (2009)), growth by collision-coalescence (e.g. **?**Onishi et al. (2015)), and radiative transfer through clouds (e.g. Kostinski (2001); Frankel et al. (2017)). Consequently, the magnitude of cloud droplet clustering *in situ* and in the laboratory has been a subject of intense interest for the last 25 years (see, e.g., Baker (1992); Baumgardner et al. (1993); Brenguier (1993); Borrmann et al. (1993); Shaw et al. (1998); Uhlig et al. (1998); Davis et al. (1999); Kostinski and Jameson (2000); Chaumat and Brenguier (2001); Kostinski and Shaw (2001); Pinsky and Khain (2001); Shaw et al. (2002); Shaw (2003); Marshak et al. (2005); Larsen (2006); Lehmann et al. (2007); Salazar et al. (2008); Saw et al. (2008); Small and Chuang (2008); Baker and Lawson (2010); Siebert et al. (2010); Bateson and Aliseda (2012); Larsen (2012); Saw et al. (2012b); Beals et al. (2015); Siebert et al. (2015); O'Shea et al. (2016)).

Most of the *in situ* studies cited above have utilized airplane-mounted cloud probes that report cloud particle positions in a long, thin, pencil-beam-like volume. For example, the sample volume of the forward scattering spectrometer probe has a cross-section of about $0.13$ mm$^2$ (Chaumat and Brenguier, 2001). These very thin sample volumes have required the majority



of the above investigators to treat cloud particle detections as 1-dimensional transects through a 3-dimensional medium and appeal to isotropy and spatial homogeneity to infer 3-dimensional statistical properties (see, e.g., Holtzer and Collins (2002)). Unfortunately, recent work (Larsen et al., 2014) reveals that – even under isotropic and homogeneous conditions – sampling requirements require far more data than initially suspected to reliably recreate 3-dimensional statistics from 1-dimensional

transects through a cloud.

The most direct and assumption-free way to detect cloud particle clustering is with an instrument that is capable of recording precise particle locations in all three spatial dimensions. This can be done with a holographic image of a cloud volume. Some previous holographic studies that explicitly examined three-dimensional cloud particle spatial distributions have been published (see, e.g., Conway et al. (1982); Kozikowsa et al. (1984); Brown (1989); Borrmann et al. (1993); Uhlig et al. (1998)). These

pioneering studies were often based on ground based measurements, included just a few holographic images, and resulted in somewhat conflicting findings. In most cases, the investigators in the above studies argued that holographic imaging looks like a solid approach to quantify cloud droplet clustering, but the excessive labor required to reconstruct the particle positions from a holographic image made the use of holographic instruments impractical for a large scale study at the time.

Fortunately, both computational and measurement hardware capabilities, as well as analysis methods, have improved im-

mensely over the last decade, finally bringing holography to a fully digital state that allows for data collection and processing over entire field projects (e.g. Fugal and Shaw (2009); Beals et al. (2015); O'Shea et al. (2016); Glienke et al. (2017); Schlenczek et al. (2017)). For example, the ability to analyze three-dimensional clustering in digital holograms has already been used to identify and eliminate particle shattering effects (e.g Fugal and Shaw (2009); Jackson et al. (2014); O'Shea et al. (2016)) or to identify regions of strong entrainment and inhomogeneous mixing (Beals et al., 2015). These new holographic instru-

ments should also allow for direct characterization of cloud droplet clustering in three dimensions while obtaining sufficient data to yield unambiguous results.

There are many different mathematical tools utilized to characterize the droplet clustering among cloud droplets, each with their own strengths and weaknesses (see, e.g., Baker (1992); Kostinski and Jameson (2000); Shaw et al. (2002); Shaw (2003); Marshak et al. (2005); Baker and Lawson (2010); Larsen (2012); Monchaux et al. (2012)). Although arguments can be made

for any number of these tools, this study focuses on the radial distribution function (rdf or $g(r)$) because (i) it is a direct scale-localized measure of deviation from perfect spatial randomness, (ii) it is directly related to variances and means through the correlation-fluctuation theorem, (iii) many numerical and theoretical discussions about particle clustering are explicitly presented in terms of the radial distribution function (see, e.g., (Balkovsky et al., 2001; Holtzer and Collins, 2002; Collins and Keswani, 2004; Chun et al., 2005; Salazar et al., 2008; Saw et al., 2008; Zaichik and Alipchenkov, 2009; Monchaux et al.,

2012; Saw et al., 2012a; Larsen et al., 2014)) and (iv) most other common methods of characterizing cloud droplet clustering can be derived from or quantitatively related to a measurement of the radial distribution function (Landau and Lifshitz, 1980; Kostinski and Jameson, 2000; Shaw et al., 2002; Larsen, 2006, 2012).

Although calculation of the three-dimensional radial distribution function from experimentally measured particle position data should be possible, properly accounting for the effects of the edges of the measurement volume can be tricky (Ripley,

1982). (This is in contrast to the much more straightforward calculation of the radial distribution function in numerical simu-



lation domains with periodic boundary conditions (e.g. Reade and Collins (2000); Wang et al. (2000)).) The most commonly utilized method does not make optimal use of the available data and is unable to estimate the radial distribution function at spatial scales larger than approximately one-half the smallest length-scale defining the measurement volume $L$. The new method developed in this manuscript removes both of these limitations.

5    The remainder of this manuscript (i) re-introduces the radial distribution function, (ii) presents the methods typically used to estimate the radial distribution function in different experimental and numerical contexts, (iii) outlines the challenges in utilizing these existing methods for experimental data from modern digital holographic images, (iv) presents and tests a new numerical method to calculate the radial distribution function under realistic experimental conditions, and (v) applies this method to real data taken by a digital holographic instrument in a cloud chamber.

## 2   INTRODUCTION TO THE RADIAL DISTRIBUTION FUNCTION

The radial distribution function is one of the most widely used approaches for characterizing particle clustering in turbulent flows (Monchaux et al., 2012), and is also currently widely used in a variety of other fields including stochastic geometry (e.g. Stoyan et al. (1995)), astrophysics (e.g. Martinez and Saar (2001)), granular media (e.g. Lee and Seong (2016)), crystallography (e.g. Cherkas and Cherkas (2016)), and plasma physics (e.g. Erimbetova et al. (2013)). The ideas behind its use go back at least a century (e.g. Ornstein and Zernike (1914)), and its wide use permits a large number of different conceptual and notational conventions.

Here, we draw on the introduction given in Landau and Lifshitz (1980) which introduces a similar quantity (the pair-correlation function) in terms of the spatial correlation of density fluctuations (section 116). Let two small disjoint volumes $dV_1$ and $dV_2$ be separated in a statistically homogeneous domain where the mean number density of particles is given by $\bar{n} = N/V$. The volumes are small enough that detection of more than one particle in $dV$ is vanishingly small. If the spatial separation between the centers of $dV_1$ and $dV_2$ is $r$, then the probability that both volumes contain a particle can be written:

20

$$p_{(1,2)}(r) = (\bar{n}\,dV_1)(\bar{n}\,dV_2)g(r) \tag{1}$$

where $g(r)$ is the radial distribution function. For perfectly random media with no spatial correlations, $p_{(1,2)}(r) = (\bar{n})^2 dV_1 dV_2$ and, thus, $g(r) = 1 \forall r$. If mutual detection in $dV_1$ and $dV_2$ is impossible at separation $r_\circ$ (due to, say, excluded volume effects) then $g(r_\circ) = 0$. If $g(r)$ exceeds unity, this indicates that there is an enhanced probability of particle-separation at scale $r$.

25

## 3   COMPUTING THE RADIAL DISTRIBUTION FUNCTION

In contexts where the spatial coordinates of each member of a population of particles are resolved, the radial distribution function at scale $r_\circ$ can be computed via calculation of



$$g(r) = \frac{\text{observed number of particle pair centers separated by } (r_\circ - \delta r < r < r_\circ + \delta r)}{\text{number of expected particle pair centers separated by } (r_\circ - \delta r < r < r_\circ + \delta r) \text{ in a Poisson distribution}} \qquad (2)$$

where the Poisson distribution has the same total number of particles and volume as the observed system. This can be rewritten algorithmically in any number of dimensions (Saw et al., 2012a) as

$$g(r) = \sum_{i=1}^{N} \frac{\psi_i(r)/N}{(N-1)\left(\frac{dV_r}{V}\right)} \qquad (3)$$

where $\psi_i(r)$ is a count of the number of particles having their centers a distance between $r - \delta r$ and $r + \delta r$ from the center of the $i$th particle in the measurement volume. $N$ is the total number of particles in the measurement volume, $V$ is the measurement volume, and $dV_r$ is the volume of the generalized n-dimensional shell between radii $r - \delta r$ and $r + \delta r$.

### 3.1 COMPUTING THE RADIAL DISTRIBUTION FUNCTION IN 1 DIMENSION

Calculation of $g(r)$ (or its related quantity, the pair-correlation function $\eta(r) \equiv g(r) - 1$) has been frequently done on *in situ* cloud particle data. Typically, a time-series of particle detections is converted to spatial positions along a line utilizing the Taylor frozen-field hypothesis (Saw et al., 2012b). Then, equation 3 is modified to:

$$g_{1D}(r_\circ) = \frac{N_p(r_\circ)}{[N_{\text{in}}(r_\circ) + \frac{1}{2}N_{\text{ex}}(r_\circ)]2(\delta r)(N-1)/L} \qquad (4)$$

where detected particle centers are located between 0 and $L$, $N_p(r_\circ)$ is the number of observed particle centers separated by $r - \delta r < r_\circ < r + \delta r$, $N_{\text{in}}(r_\circ)$ is the number of observed particles detected between $r_\circ$ and $L - r_\circ$, and $N_{\text{ex}}(r_\circ)$ are the number of observed particles detected between 0 and $r_\circ$ plus the number of observed particles detected between $L - r_\circ$ and $L$. The factor of $1/2$ multiplying $N_{\text{ex}}(r_\circ)$ is sufficient to account for the edges of the sample volume in the 1-dimensional case. Since typically $r_\circ << L$, this is often simplified to

$$g_{1D}(r_\circ) \approx \frac{N_p(r_\circ)}{2N(N-1)(\delta r)/L}. \qquad (5)$$

The above formula has been used in most previous experimental studies computing the radial distribution functions for cloud droplets. In principle, this result can then be used to estimate the 3-dimensional radial distribution function following the method outlined in Holtzer and Collins (2002), though the assumptions of statistical homogeneity over the tens to hundreds of kilometers required for obtaining a statistically significant results may be questionable (Larsen et al., 2014).

### 3.2 COMPUTING THE RADIAL DISTRIBUTION FUNCTION IN MULTIPLE DIMENSIONS WITH PERIODIC BOUNDARY CONDITIONS

The three-dimensional radial distribution function *can* be explicitly computed for cloud droplets in drop-resolving direct numerical simulations. In this context, $g(r)$ can be directly evaluated from equation 3 without any modification. The saving grace





that allows computation of the rdf in these scenarios is that the numerical simulations utilize periodic boundary conditions – which extend to the computation of the rdf itself.

When searching for another cloud droplet separated by scale $r_\circ - \delta r < r < r_\circ + \delta r$, any part of the "search domain" outside of the simulation volume can be wrapped back around through the other side of the computational domain. Since the underlying

simulation typically applies this same wrapping boundary condition to resolve particle-fluid and particle-particle interactions, it is consistent with the physics of the simulation to search for particle-pairs across the boundaries as well. A cartoon of this process (shown in two dimensions) can be viewed in the left panel of figure 1.

### 3.3   COMPUTING THE RADIAL DISTRIBUTION FUNCTION IN MULTIPLE DIMENSIONS WITHOUT PERIODIC BOUNDARY CONDITIONS

Unfortunately, the technique described in the left-hand panel of figure 1 is not appropriate for most experimental contexts; detected particles on opposite sides of the sample volume do not "know" about each other in the same way that simulations applying periodic boundary conditions do.

The simplest possible solution, albeit the most drastic, in trying to estimate the radial distribution function for finite experimental volumes is to ignore these edge effects entirely. For the cartoon in the left panel of figure 1, this would be to merely

count the 1 particle detected in the yellow ring and do nothing to account for the blue area at all. Unfortunately, this will cause a computational estimate of $g(r)$ to artificially deviate from unity; actual cloud droplets may exist in the blue area and need to be counted in order to prevent artificial underestimation of $\psi_i(r)$ and therefore underestimation of $g(r)$.

Much like in the 1-dimensional case, the effects of the edges sometimes can be small enough to make this a minor concern. When the scale of interest $r_\circ$ is much less than the smallest dimension of the sample volume ($L$), relatively few particles inside

the sample volume will have their $n$-dimensional spherical shells exit the interior of the measurement volume. Unfortunately, however, (i) experimental conditions for cloud droplets will require estimation of $g(r_\circ)$ for $r_\circ$ approaching $L$ in order to maximize the evaluated range of $r$, and (ii) the problem gets more prevalent in higher dimensions and in larger aspect ratios since a larger fraction of the measurement volume is found close to the boundaries.

As noted earlier, this is a problem that has received attention for at least 35 years (Ripley, 1982). Perhaps the most common

way to deal with these finite-volume effects is described as "minus-sampling" on p. 133 of Stoyan et al. (1995) and illustrated in the right half of figure 1. Briefly, one defines a "guard area" within but along the outermost edges of the sampling volume. Particles inside this guard area are not considered part of the actual sample volume $V$, but *are* used to find pairs for particles within the central (non-guard) part of the measurement volume. Note that $R$ can be either fixed or change with the scale of interest (set $R = r_\circ$ when computing $g(r_\circ)$.). The guard-area approach does give an unbiased estimator for $g(r)$, but makes

sub-optimal use of the data. Two particles within the sample volume could be separated by scale $r_\circ - \delta r < r < r_\circ + \delta r$ but end up not contributing to the observation, due to the fact that both particles would be in the guard-area. Much of the data is lost when using such approaches.

Figure 2 shows another cartoon that demonstrates how limiting the guard area approach can be in different contexts. Here, $R = r$ is only slightly smaller than $L/2$. The "inner" particles that contribute to the sum in equation 3 are only the 5 particles





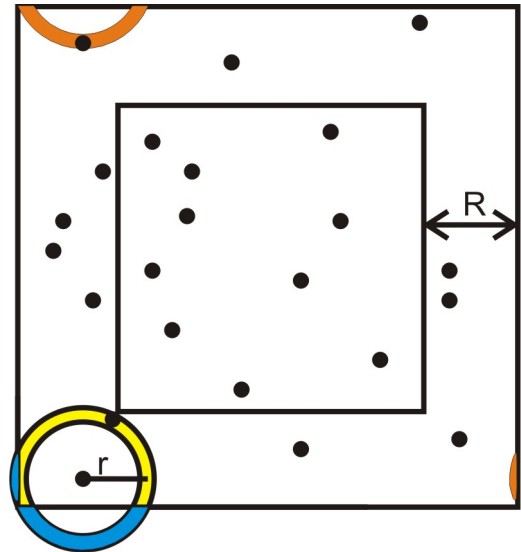
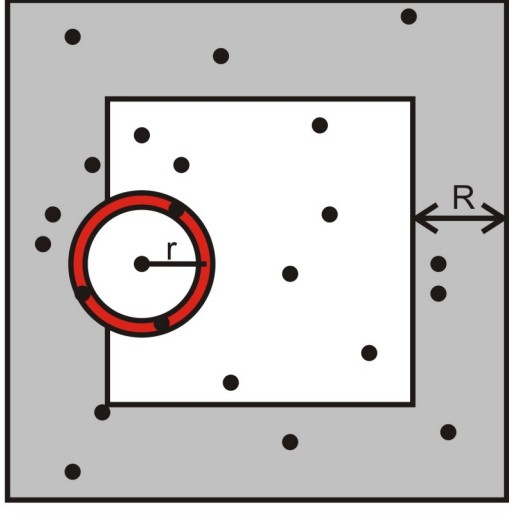

**Figure 1.** A two-dimensional cartoon of the different ways of traditionally dealing with domain edges when computing the radial distribution function. The left panel shows the need for periodic boundary conditions when the entire experimental domain is used. Some of the circular shell surrounding the particle in the lower left leaves the measurement volume (blue portion). When the data comes from a direct numerical simulation, there are no issues in wrapping this volume around to the upper left and lower right corners (to the orange regions), in this case finding an additional particle pair in the upper left. In an actual experiment, however, it is a mistake to argue that the particle in the lower left is correlated to the particle in the upper left at a length scale of $r$ since they are in fundamentally different parts of the flow (i.e. any correlation that does exist is for length scale equal to the non-periodic distance between the particles usually much greater than $r$). The right panel shows how this obstacle is typically overcome by using a "guard area". The entire measurement volume still corresponds to both the grey and white boxed areas, but now a guard-area of size $R$ is marked off along the periphery of the sampling volume. Though all particles in both the white and grey areas are detected, only particles within the white square are used in the sum defining $g(r)$; for the particle highlighted, there are 3 particles between $r - \delta r$ and $r + \delta r$ that are counted. By only counting pairs originating from particles in the white area, it is assured that all particles distance $r < R$ from the particles in the white area are counted. This is a valid approach for finite-volume cloud measurements, but it imposes a trade-off: either most of the volume can be used, but with severely limited maximum $r$, or the available sample volume is severely reduced in order to accommodate a maximum $r$ that is of the same order as the sample volume linear dimensions. Typically as large a range of $r$ as possible is desired (e.g. in order to have enough scale range to reliably identify power-law exponents), but the associated reduction in available sample volume makes the method quite susceptible to sampling fluctuations.

shown inside the central white rectangle. This problem is even worse in 3d, and the aspect ratio shown here is not unrealistic. In scenarios where the entire measurement volume contains only a few hundred to a few thousand particles, sampling considerations make use of the guard area technique prohibitively limiting.

   Here, we introduce an alternative edge-correction strategy inspired by Ripley (1976, 1977) that we call the "effective-
5  volume" radial distribution function method. This approach does not rely on the use of a guard-area and allows all retained





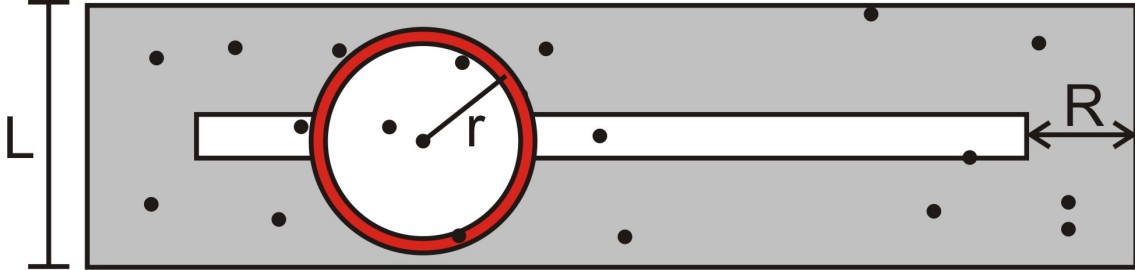

**Figure 2.** Another cartoon of the guard-area technique used to estimate the radial distribution function. Note that the fraction of the particles contributing to the sum in equation 3 decreases as the aspect ratio increases, and no estimate of the rdf can be made for any distance larger than half the shortest dimension of the sample volume (when $r \geq L/2$, no "inner" region remains).

particles to contribute to the computation of the radial distribution function. We start from a refined expression for the radial distribution function for length-scale $r_j$:

$$g(r_j) = \sum_{i=1}^{N} \frac{\psi_i(r_j)/N}{(N-1)\left(\frac{\mathrm{d}V_{r_{i,j}}}{V}\right)}. \tag{6}$$

This is very similar to equation 3, except we have made the computationally-motivated step of discretizing the set of distances
$r_j$ and defined a quantity $\mathrm{d}V_{r_{i,j}}$ which is defined as the portion of the volume with radius between $r_j - (\delta r)_j$ and $r_j + (\delta r)_j$ centered on the $i$th particle that resides within the measurement volume $V$. (For example, in the left panel of figure 1, $\mathrm{d}V_{r_{i,j}}$ for the highlighted particle would be calculated as that area corresponding to the yellow region). This depends not only on $r_j$ and $(\delta r)_j$, but also on the position of the $i$th particle. Thus, within this method, the denominator is not a constant and must be explicitly calculated particle by particle.

The challenging part of the method is to find $\mathrm{d}V_{r_{i,j}}$; all other parts of the numerical method are the same as have been used elsewhere. Although potentially inelegant, $\mathrm{d}V_{r_{i,j}}$ can be found for a wide variety of measurement geometries by generating a measurement geometry-dependent look-up table. This can be accomplished by computing values of $\mathrm{d}V_{r_{i,j}}$ in a dense grid of possible positions of each detected particle and at each desired distance $r_j$. Since for convex volumes it is empirically found that $\mathrm{d}V_{r_{i,j}}$ is relatively smooth over the measurement domain, one can then assign $\mathrm{d}V_{r_{i,j}}$ for the $i$th particle at the $j$th radial
distance by utilizing the closest look-up table stored grid-point to the actual particle position.

There are multiple ways to generate the proposed look-up table. In this study, we have populated the interior of the measurement volume with a regular dense grid with grid-spacing $s$. Then, for each grid-point and for each scale of interest $r_j$, the number of other grid points contained in a shell with inner and outer radii $r_j - (\delta r)_j$ and $r_j + (\delta r)_j$ are counted. This is then compared to the number of grid points that would be contained in a shell of the same volume within an infinite grid with the
same grid-spacing $s$. The ratio of these two counts is then multiplied by the true volume of the shell $\mathrm{d}V_j$ to give $\mathrm{d}V_{r_{i,j}}$. This method allows for reliable estimation of $\mathrm{d}V_{r_{i,j}}$ without having to mathematically calculate the quantity analytically, which would require rather lengthy treatments of possible boundary/shell intersection geometries (especially in 3-dimensions).





Conceptually, the algorithm uses the $i$th term in the sum in equation 6 to find an appropriately weighted contribution to $g(r)$ from the $i$th particle; when summed over all particles in the measurement volume, the expression gives an estimator for $g(r)$ that accounts for edge effects. This weighting factor appears in the denominator and is based on a term that depends on how close the $i$th particle is to the edge of the measurement volume; if the particle in question is within $r_j$ of the edge of the

measurement volume, only the portion of the $n$-dimensional spherical shell $dV$ that still lies within the measurement domain is used.

The effective volume method allows for any investigator-chosen values of $r_j$ and $(\delta r)_j$, allows for as fine of a tesselation of the measurement volume as desired for precision in the look-up table, and can be used even for $r_j > L$. Once the look-up table is generated it can be applied to all data in a data set, assuming the instrument measurement volume shape and size is constant.

It should be further noted that symmetry in the measurement volume shape can be used to reduce the number of normalization volumes that need to be calculated. For example, in a rectangular parallelepiped only one octant (corner) of the measurement volume needs to be in the look-up table. More explicit detail on how to implement this method is presented in the appendix.

## 4   TESTING THE EFFECTIVE VOLUME METHOD

The effective volume method described above was implemented for two different geometries – a cubical geometry (to allow

for useful comparisons to the well-known and frequently utilized guard-area technique) as well as in an applied geometry to match a real instrument. For each geometry, we present two tests: a homogeneous Poisson distribution and a Matérn Cluster Process.

A homogeneous Poisson distribution is the gold standard of spatial randomness. Within a homogeneous Poisson distribution, all particles are placed independently with a spatial density function uniform over the measurement domain. By construction,

$g(r) = 1 \forall r$ within a volume with particles distributed according to a homogeneous Poisson distribution.

A Matèrn Cluster Process (see, e.g., Stoyan et al. (1995); Martinez and Saar (2001); Schabenberger and Goway (2005); Larsen (2012)) is commonly used in stochastic geometry because it is (i) statistically homogeneous, (ii) easy to simulate in any number of spatial dimensions, and (iii) has a known closed-form expression for its radial distribution function.

### 4.1   SIMULATIONS IN A CUBE

Both distributions described above were simulated within a unit cube with both guard area and effective volume computation methods.

For the guard area computation method, a fixed $R = 0.1$ was used around the outside edges of the cube. This should allow for unbiased estimation for $g(r)$ when $r < 0.1$, but it is expected that $g(r > 0.1)$ will underestimate the true values.

The effective volume computation method was computed by creating a look-up table for a cubical volume. Due to the

symmetry of the volume, only one octant of the cube had to be included in the look-up table. To minimize the size of the look-up table required, the density of the tesselation of the cubical measurement volume was varied depending on the distance to the boundary – with points near the boundary having the densest collection of look-up table entries.




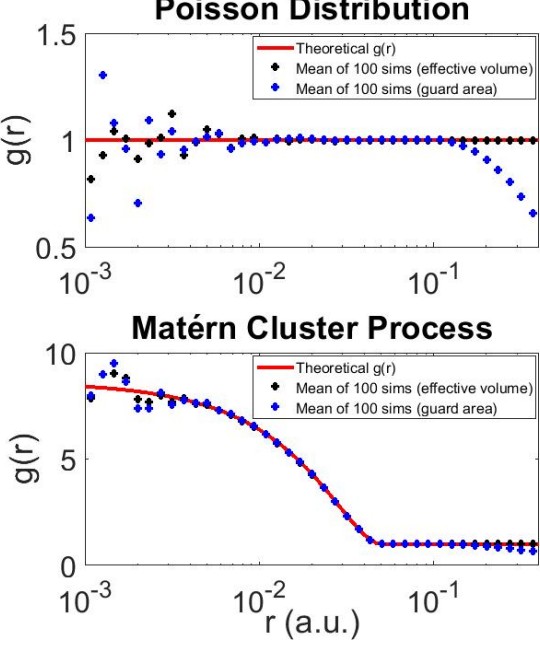

**Figure 3.** First verification of the method for calculating $g(r)$ described in the main text. Here, theoretical curves of $g(r)$ are compared both to the effective volume method and the guard-area method (using a fixed guard area of 0.1 times the side-length of the cube). In the top panel, 100 different 10,000 particle Poisson distributions were created. Deviations from $g(r) = 1$ in both methods are observed at small $r$ due to sampling fluctuations. The bottom panel shows similar results from 100 simulations of a Matérn Cluster Process (with a mean of 10,000 total particles and a cluster length of 0.025). Note that in both panels the guard area approach begins to fail as expected for $r > 0.1$.

One hundred simulations of each volume were averaged together and the results are displayed in figure 3. Except for the smallest scales (where sampling variability is still non-negligible, even after a total of 100 simulations), agreement between the effective volume method and the theoretical $g(r)$ curves expected is excellent and comparable to the values observed for the more commonly used "guard-area" approach.

5     In this case, the "guard-area" approach involves summing over less than half (on average 48.8%) of particles in the measurement volume, which can help to explain the larger scatter of observed $g(r)$ for small $r$ using this approach. Note also the deviation from the theoretical $g(r)$ curve in both tests for the guard-area approach for $r > 0.1$, consistent with pushing the approach beyond its domain of applicability.

## 4.2   CASE STUDY: THE HOLOGRAPHIC DETECTOR FOR CLOUDS (HOLODEC)

10   Although the effective volume approach introduced here performs approximately as well as the more traditional guard area approach in cubical volumes, the development of the new method was primarily motivated by a desire to estimate the radial distribution function in contexts where the guard-area approach will not work. As noted previously, when estimates of $g(r)$




are desired for $r \gtrsim L/2$ and/or the aspect ratio of the measurement volume deviates substantially from unity, the guard-area approach becomes ineffective.

An example of an instrument that is subject to these limitations and is relevant for studying cloud particle clustering is the holographic detector for clouds (HOLODEC).

### 4.2.1  INTRODUCTION TO HOLODEC

HOLODEC is an in-line digital holography instrument explicitly designed to explore cloud microstructure (Fugal et al., 2004; Fugal and Shaw, 2009; Spuler and Fugal, 2011). The instrument has previously been used to examine drop size distribution and liquid water content fluctuations on the centimeter scale (Beals et al., 2015), and the behavior of the instrument has been validated by comparison to co-collected cloud droplet probe (CDP) and 2DC optical array probe data in different parts of the particle size domain (Glienke et al., 2017).

A processed HOLODEC hologram reports droplet positions in a volume that is approximately 1 cm x 1 cm x 15.8 cm with sensitivity to all droplets with sizes greater than about 6.5 $\mu$m. The positional uncertainty for each drop is approximately 10 $\mu$m along the short sides of the sample volume and about 100 $\mu$m along the longer side (Yang et al., 2005).

A 2-dimensional cartoon of the HOLODEC sample volume is shown in figure 4. Although particles out to 158 mm (or further) from the hologram plane are potentially visible, the optical windows 14 mm and 158 mm from the hologram plane limit the air-exposed field of view to the approximately 14 cm distance between the windows. Additionally, the spatial domain of instrumental sensitivity is not a perfect parallelepiped. Preliminary analyses of data suggests that there may be some decreased sensitivity near the edges of the sample volume, and – when mounted on an aircraft – drops can be created by fragmentation near the optical windows. Consequently, to ensure data fidelity when used with real data a conservative sub-volume of each hologram is selected as the measurement volume for analysis. This sub-volume was selected to be in the central part of each hologram where the data is expected to be most reliable. Thus, the used HOLODEC sample volume is a 6 mm x 6 mm x 10 cm rectangular parallelepiped.

### 4.2.2  SIMULATIONS WITHIN THE INSTRUMENT DOMAIN

The same two tests (homogeneous Poisson distribution and Matérn Cluster Process) were simulated within the parallelepiped sample volume of the HOLODEC. A new look-up table for this geometry was generated and used to calculate $g(r)$ for each of 100 different simulations, with the mean value of $g(r)$ compared to theoretical expectations and shown in figure 5. The guard area method for calculating $g(r)$ is not shown, since it is susceptible to substantial sampling variability at all scales and cannot be used at all for any scale larger than 3 mm (the entire volume is then the guard area).

In general, the agreement between the theroretical expressions for $g(r)$ and the measured $g(r)$ is excellent, and suggests that the effective volume radial distribution function computational method should work for real data.





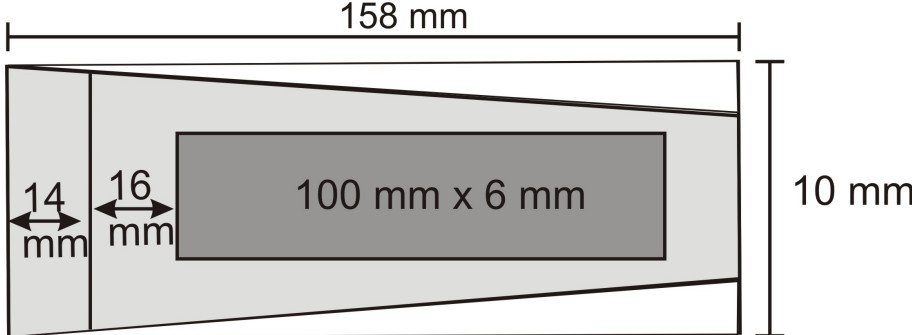

**Figure 4.** A 2-dimensional cartoon of the HOLODEC sample volume (not to scale). The leftmost vertical line in the figure indicates the hologram plane. The light grey region indicates areas of maximum sensor sensitivity (the slope of the angled lines marking the edge of the light grey region have been greatly magnified for aid in visualization). The vertical lines 14 mm and 158 mm from the left edge of the figure mark the positions of the optical windows; near these windows there is evidence of artificially generated particles due to instrument-induced particle fragmentation. The volume simulated here corresponds from the darker central grey rectangle (parallelepiped in 3d), where the instrument retains approximately uniform sensitivity, particle locations and sizes are believed to be accurate, and the number of small particles generated due to fragmentation on the instrument is believed to be negligible.

### 4.2.3 REAL DATA

To test the claim made above, a proof-of-principle analysis was completed using real HOLODEC data acquired inside a laboratory cloud chamber driven by Rayleigh-Bénard convection (Chandrakar et al., 2016; Chang et al., 2016; Chandrakar et al., 2017; Desai et al., 2018).

The radial distribution functions of the 8 holograms with the largest numbers of detected drops are shown in figure 6. For these 8 holograms, there were an average of about 185 cloud drops/cubic centimeter within the measurement domain (which is reasonable compared to naturally-occurring clouds).

These single-hologram results are noisy due to the sampling uncertainty,(especially for the smallest spatial scales), but it is clear that there is systematic deviation from $g(r) = 1$ for $r \lesssim 1$mm. Quantitative comparisons to theoretical expectation may

be too much to ask from such a limited data-set, but it is promising to note that the observed increase in $g(r)$ with decreasing length scale $r$ within the dissipative range is consistent with expectations for inertial clustering of particles in a turbulent flow (Reade and Collins, 2000; Ayala et al., 2008; Saw et al., 2012a).

## 5   Conclusions

Understanding the effects of cloud particle clustering on microphysical processes requires reliable estimation of the three-
dimensional radial distribution function. Previous studies have obtained this information by utilizing one-dimensional measurements of cloud particle positions to infer scale-dependent clustering, but these methods have been shown to carry large un-





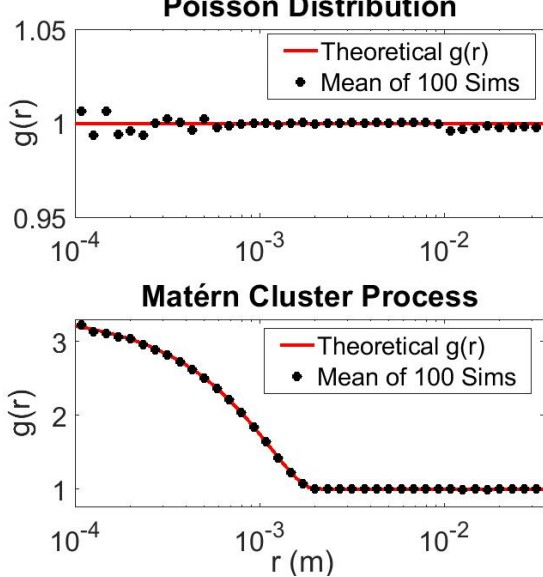

**Figure 5.** Verification that the effective volume method for calculating $g(r)$ as described in the main text works for non-cubical sample volumes with realistic aspect ratios. In the top panel, 100 different simulations of a Poisson distribution (perfect spatial randomness) were created by placing 10,000 particles within a sample volume with the same dimensions as the HOLODEC sample volume. The mean of the 100 simulations agrees very well with the theoretical $g(r) = 1$ curve. An unrealistically large number of particles was used for each simulation in order to minimize the sampling concerns. Note that though the agreement is very good, the most pronounced deviations from the theoretical curve still occur as expected at small spatial scales. In the bottom panel, 100 different simulations of a Matérn Cluster Process were generated and compared to the known theoretical expression (see, e.g., Larsen et al. (2014)). Clearly, agreement between the mean of the simulations and the theoretical curve is excellent.

certainties. To minimize these uncertainties, measurement of the radial distribution function for *in situ* data in three-dimensions is desired.

Comparing measurements with theory and numerical simulation relies on estimating $g(r)$ over a wide range of spatial scales and making optimal use of the measured data to combat sampling uncertainties. Because the aspect ratios of the holographic instruments designed to explore three-dimensional cloud microstructure are large and the radial distribution function must be estimated on scales exceeding the smallest dimension of the measurement volume, standard computational methods that use spatial information to estimate the radial distribution function are not adequate.

Here, a new method was introduced that explicitly considers each particle's position within the measurement volume in the radial distribution function computation. This method allows for calculating the radial distribution function for scales larger than the shortest physical dimension of the measurement volume, and makes more optimal use of the measured data. This "effective-volume" method was tested in two different geometries, compared to standard computational methods with simulated data in a unit cube, and validated in a more realistic sampling scenario.





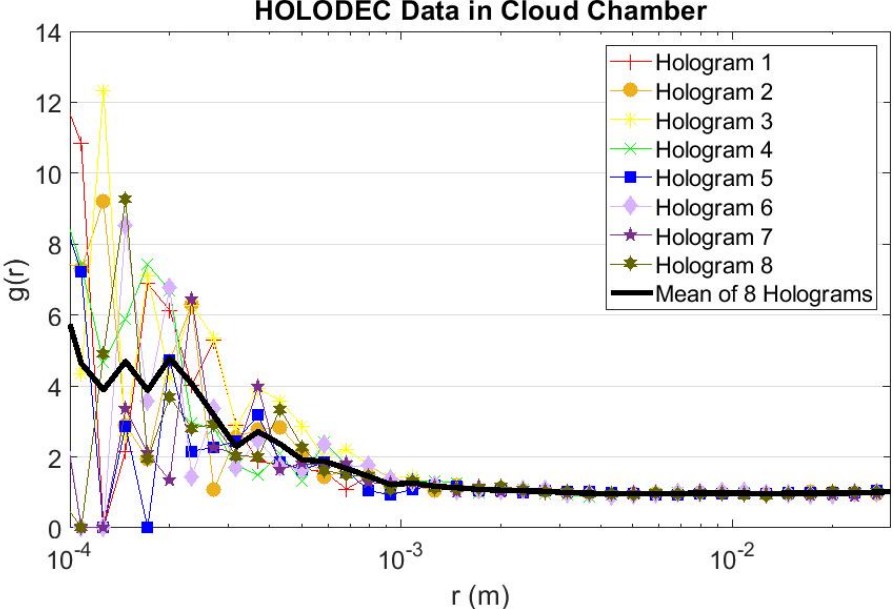

**Figure 6.** The measured radial distribution functions for 8 different holograms and their mean for HOLODEC data taken in the cloud chamber. Clearly sampling variability is still pronounced at small spatial scales, but scatter substantially diminishes with increasing spatial scale. Some evidence of scale-dependent clustering is present.

Preliminary results confirm that use of the effective volume method should enable the use of airborne digital holography data to compute *in situ* three-dimensional radial distribution functions for cloud droplets.

*Data availability.* The HOLODEC data associated with the analysis in section 4.2.3 is available by request from the authors.

**Appendix A: Basic Structure of Codes to Use Effective-Volume Method**

5  The effective volume method to calculate the radial distribution function relies on two codes – one to generate a look-up table for the measurement volume, and another to use the look-up table and data to compute the radial distribution function. This appendix outlines the basic structure utilized for each of these codes.

**A1   Generating the Look-Up Table**

Required inputs from the user: Physical domain of sample volume, set of radii $r_j$ and associated ranges $(\delta r)_j$, and grid
10  tessellation scale $s$ (as small as computationally feasible).

1. Tessellate the interior of the sample volume domain at scale $s$, giving a total of $M$ grid points.



2. For each radius $r_j$, and for each grid point $i = 1 : M$, compute the number of grid points inside the $n$-dimensional shell centered on the $i$th grid point with inner and outer radii $r_j - (\delta r)_j$ and $r_j + (\delta r)_j$, respectively. Store the result as $a(i, j)$.

3. Tessellate an $n$-dimensional cube at scale $s$ with side lengths $2\left[\max\left(r_j + (\delta r_j)\right)\right]$.

4. For each radius $r_j$, compute the number of grid points inside the $n$-dimensional shell centered on the center of the $n$-dimensional cube with inner and outer radii $r_j - (\delta r)_j$ and $r_j + (\delta r)_j$, respectively. Store the result as $b(j)$.

5. Compute the factor $\mathrm{norm}(i, j) = a(i, j)/b(j)$.

## A2    Using the Look-Up Table and Data to Compute a Radial Distribution Function

Required inputs include the same set of inputs utilized to generate the look-up table, and $N$ different $n$-dimensional particle positions.

1. Load look-up table

2. For each radius $r_j$, calculate the volume of the $n$-dimensional sphere of the shell between radii $r_j - (\delta r)_j$ and $r_j + (\delta r)_j$. Store the results as $dV(j)$.

3. For each particle $k = 1 : N$ and for each radius $r_j$

   (a) Count the number of other particles that are between $r_j - (\delta r)_j$ and $r_j + (\delta r)_j$ from the $k$th particle and store the result as $\psi(k, j)$.

   (b) Identify the closest entry in the look-up table $i$ to the associated position of the $k$th particle, store as $p$.

   (c) Assign $dVr(k, j) = dV(j) \cdot \mathrm{norm}(p, j)$ .

   (d) Use $dVr(k, j)$ and $\psi(k, j)$ to compute the $k$th term of the sum for $g(r)$ following equation 6 to give $g(k, j)$.

4. Compute and return $g(j) = \sum_{k=1}^{N} g(k, j)$

*Competing interests.* The authors declare that there is no conflict of interest.

*Acknowledgements.* This work was supported by the U.S. National Science Foundation through grants AGS-1532977 (MLL) and AGS-1623429 (RAS). Special thanks to Alexander Kostinski, Susanne Glienke, and Neel Desai for helpful discussions and help with accessing and interpreting the HOLODEC data from the Π Chamber.



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
