# Peer review of "A Method for Computing the Three-Dimensional Radial Distribution Function of Cloud Particles from Holographic Images"

_Atmospheric Measurement Techniques, 2018_

## Referee Comment (RC1) · Anonymous Referee #2 · 24 Apr 2018

I find little to argue with in this paper at least within its well-defined restricted domain. Potentially it offers a wonderful way of observing a very difficult but important quantity to measure in clouds, namely the radial pair correlation function using data in three dimensions. As far as I can tell, it is still not possible to measure the three dimensional pair correlation function. Nevertheless, this work still provides a potential path for measuring an important function and, as such, is worthy of publication.

There are places, however, where I would like to see the article strengthened, most notably near the end. While the authors find Fig.6 encouraging, I find it rather dis-

couraging given the great uncertainty in g(r) in the sub-millimeter radial distances. Is it even possible to really measure g(r) in this range? What kind of position accuracy would be required to make such measurements and are they really achievable using aircraft observations? If it takes, say, 50 microseconds per hologram and an aircraft is flying at 100 m/s then the displacement between successive holograms would be 5 mm. If, say, as Fig.6 suggests, you would like on the order of 10 holograms, the aircraft would have moved 50 mm or 0.5 cm. How would that affect the calculated g(r)? You are certainly not looking at all the same particles so that the uncertainties would presumably increase. Along the same line, how many holograms do you think you would need to measure g(r) to some desired degree of accuracy in the sub-millimeter range even if conditions were stationary? While Fig.6 illustrates the variability, the addition of a plot of the relative uncertainty would also be appreciated. Is the noise white?

As referenced in the paper, while others have made concentration measurements using the HOLODEC instrument, for example, it is a wholly more demanding challenge to measure g(r) with reliable accuracy in actual clouds using aircraft measurements. This challenge should at least be acknowledged and discussed in this work as well as how this might be addressed in real world observations.
* * *

---

## Referee Comment (RC2) · Anonymous Referee #3 · 9 May 2018

**General comments**

The authors present a novel method for calculating the radial distribution function (rdf), a quantitative descriptor of clustering of particles in a volume. Among other applications, the rdf is used in the cloud microphysics community to inform process rates (i.e. as one component of turbulent collision-coalescence kernels) and diagnose mixing state. Other methods for calculating the rdf either assume an infinite domain (i.e. in direct numerical simulations of cloud drops with periodic boundary conditions) or rely on questionable assumptions to derive the rdf from effectively 1-dimensional *in*

*situ* cloud probe observations. Given that one of the authors has been a key player in the development of a probe that can measure a truly 3d cloud volume, it makes sense that the derivation of the rdf should be revisited. That said, I am not convinced that this study will find broad application beyond the small community of digital holography observationalists.

The derivation of the effective volume rdf is a rather intuitive solution. While the algorithm for computing it may be "inelegant" (authors' description), barring the development of new instrumentation with cavernous sampling volume, complex geometry and/or extremely fine pixel resolution, the brute force method of computing volume normalization factors seems sufficient. It is unfortunate that only 8 holograms were analyzed, and I would ask that the authors comment on whether there exists a threshold concentration below which *g(r)* cannot be accurately computed. Holograms with lower concentration will likely increase the "noise floor" length scale where *g(r)* begins to deviate from unity but increasing the number of samples may offset this effect. If there does exist a low concentration limit, does that imply that this analysis is only suitable for a small subset of relatively high drop concentration environments? I can envision clustering properties having some dependence on concentration, especially in light of other work using the $\Pi$ Chamber (e.g. Chandrakar *et al.*, 2016; Desai *et al.*, 2018). Finally, it is unclear to me how application of this analysis to *in situ* measurements furthers the development of microphysical parameterizations in the absence of collocated measurements of turbulence intensity or supersaturation state, and I invite the authors to expand on their vision for how this may be accomplished. These concerns are all relatively minor though, and I recommend this study for publication in AMT.

**Specific comments**

Comments are given as "page X, line/figure Y" and are listed in order from beginning to end of the manuscript.

- P. 6, Fig. 1 caption: Incorporate the narrative portion of this caption into the

main text. The information given here is important, and reading it from the figure caption makes it more difficult to connect it with what's happening in the text (i.e. 4th paragraph of section 3.3).

- P. 9, Fig. 3: It's difficult to differentiate the blue and black markers unless the figure is magnified. Consider choosing more strongly contrasting colors or using different markers. Also, I assume "a. u." stands for arbitrary units but it took me a while to figure this out – please define this in the caption.

- P. 10, L. 18-19: Is there any study you can cite re: fragmentation near the optical windows? Is this a problem for liquid, ice, or all particles?

- P. 11, L. 9-10: I agree that a quantitative comparison with the small sample here would not be appropriate, but is it worth adding a theoretical Matérn process to Fig. 6 for qualitative comparison with the mean curve? Disregard this comment if adding a theoretical curve distracts from the point you're trying to make.

- P. 11, L. 16: Is the issue that there are large uncertainties in 1d *in situ* results, or rather that it's unclear from a theoretical perspective whether they are extensible to 3 dimensions given the pile of underlying assumptions? I fully agree with the first full sentence of P. 12 that measuring the 3d rdf is highly desirable, but I don't think uncertainty should be the focus unless you can quantify how much it is reduced by increasing dimensionality. As I understand, the point is that your method requires fewer assumptions be made and there is greater consistency between measurement and application of the rdf.

- P. 12, Fig. 5 caption: Same as Fig. 1 caption - you are using the caption to communicate information that belongs in the main text.

**Technical corrections**

- P. 1, L. 12: Is the bold-faced "**?**" next to Onishi et al. (2015) a missing reference?

- P. 2, L. 28 & 30: Extra set of parentheses surrounding reference list

- P. 3, L. 1: Extra right parenthesis at end of sentence.

- P. 4, L. 6: "...in the measurement volume.  $N$ is the..." – comma instead of a period.

- P. 4, L. 14: "...and $N_{ex}(r_{\circ})$ are the number..." – should be "is" instead of "are"

- P. 11, Fig. 4 caption: "The volume simulated here corresponds from..." – should be "corresponds to"

- P. 14, L. 11: "calculate the volume of the $n$-dimensional sphere of the shell" – I'm confused by this, did you mean to say "sphere of the shell?"

- P. 14, L. 19: Is there a factor of $1/N$ missing from the equation or is it just a straight sum?

---

## Author Comment (AC1) · 21 Jun 2018

We thank the reviewers for their many constructive comments and were pleased to see that both reviewers stated that they found our manuscript worthy of publication.

Below we address individual comments from the reviews, but here we address a general question raised by both reviewers concerning the practical applicability of the described approach to holographic data from real atmospheric clouds. We can rephrase it slightly by referring to our figure 6 – showing sample data from a cloud chamber –

which clearly shows that hologram to hologram variability is significant at small spatial scales; it is therefore completely justified to be concerned as to whether the algorithm introduced here has any practical utility for *in situ* cloud measurements. The central question is therefore: does sampling variability associated with estimating $g(r)$ in real clouds prevent us from obtaining meaningful information from real clouds? The answer to this question is that reliable three-dimensional estimates of $g(r)$ can be obtained for *in situ* cloud data if an adequate number of holograms can be recorded under statistically uniform conditions. Meeting the "adequate" requirement depends very strongly on at least two factors: (i) the instrument that is being used for the sampling, specifically, its sample volume, sample geometry, and sample rate; and (ii) the properties of the cloud being sampled, specifically, its cloud droplet number density and size distribution, and its spatial uniformity. We have added text to discuss how these factors come into play (see the text now closing section 4.2.3). Because this manuscript is focused on the algorithm for estimating $g(r)$, independent of specific instrument or data features, our sense is that a full treatment of the "adequate" requirement would be a diversion.

Logically, this work comes first. Before anyone will be convinced by our analysis of *in situ* data, they have to be convinced as to the validity of the new method of computing the 3d rdf presented here. Additionally, the algorithm presented here has possible applications outside the realm of cloud physics; the radial distribution function is used in a wide variety of disciplines from astronomy to x-ray diffraction and – to our knowledge – no study like this has been presented for finite-domain measurements anywhere, even in those communities.

The questions raised by the reviewers are absolutely critical, and in fact, they motivated this work in the first place. The role of instrument-specific sampling properties and cloud properties are central elements of another manuscript that the authors (along with other colleagues) have under review elsewhere. We feel that the work presented here stands alone in describing and validating the algorithm, and fits well within the scope of this journal.

Below, please find our summary comments associated with both public reviews of our manuscript. Each reviewer comment is given in *italics* and our responses are written in plain text.

**Reviewer # 2**

*I find little to argue with in this paper at least within its well-defined restricted domain. Potentially it offers a wonderful way of observing a very difficult but important quantity to measure in clouds, namely the radial pair correlation function using data in three dimensions. As far as I can tell, it is still not possible to measure the three dimensional pair correlation function. Nevertheless, this work still provides a potential path for measuring an important function and, as such, is worthy of publication.*

We are encouraged that this reviewer feels that the work is worthy of publication.

*There are places, however, where I would like to see the article strengthened, most notably near the end. While the authors find Fig. 6 encouraging, I find it rather discouraging given the great uncertainty in $g(r)$ in the sub-millimeter radial distances. Is it even possible to really measure $g(r)$ in this range? What kind of position accuracy would be required to make such measurements and are they really achievable using aircraft observations?*

We are encouraged by Fig. 6 primarily because – to our knowledge – it is the first time that the radial distribution function has been computed for a real holographic image of cloud-droplets. We agree with the reviewer that individual holograms – and even the collection of 8 holograms shown – are not giving us useful statistical information on scales below a millimeter.

In regards to the questions the reviewer poses – they are very important and are discussed in the opening section. We have added some discussion of this point to the

paper (section 4.2.3). Detailed discussion that requires delving into specifics of the sampling instrument and the cloud being sampled are addressed in another paper under consideration elsewhere.

*If it takes, say, 50 microseconds per hologram and an aircraft is flying at 100 m/s then the displacement between successive holograms would be 5 mm. If, say, as Fig. 6 suggests, you would like on the order of 10 holograms, the aircraft would have moved 50mm or 0.5 cm. How would that affect the calculated $g(r)$? You are certainly not looking at all the same particles so that the uncertainties would presumably increase. Along the same line, how many holograms do you think you would need to measure $g(r)$ to some desired degree of accuracy in the sub-millimeter range even if conditions were stationary? While Fig. 6 illustrates the variability, the addition of a plot of the relative uncertainty would also be appreciated. Is the noise white?*

This is addressed in the previously noted paper under consideration elsewhere. By way of preview, we have found that the reviewer's suspicions are correct; individual holographic images from HOLODEC do not give sufficient information to reveal meaningful information about spatial scales less than a cm or so in realistic drop number concentrations. However, a collection of many holograms can reduce the uncertainty related to the radial distribution function to a degree where information down to scales of $\sim 1$ mm may be achievable.

Regarding the reviewer's request about gaining sufficient data from a number of different holograms related to a flight, the good news is that the effective-volume rdf method does not necessarily require the examined volumes to be physically adjacent. Each hologram can be examined independently and then the resulting rdfs can be combined (so long as the assumption of statistical stationarity and isotropy is maintained not only within each hologram but also between holograms). Although the question of stationarity on larger spatial scales will always be a constraint, when this assumption can be granted the effective-volume rdf method allows for examination of much larger volumes while only sub-sampling a miniscule fraction of the associated cloud volume.

*As referenced in the paper, while others have made concentration measurements using the HOLODEC instrument, for example, it is a wholly more demanding challenge to measure $g(r)$ with reliable accuracy in actual clouds using aircraft measurements. This challenge should at least be acknowledged and discussed in this work as well as how this might be addressed in real world observations.*

We agree that the manuscript as initially submitted did gloss over the practical use of this tool too much. In order to respond to this concern, we have added some text that points out the practical challenges of analyzing *in situ* data.

**Reviewer # 3**

*The authors present a novel method for calculating the radial distribution function (rdf), a quantitative descriptor of clustering of particles in a volume. Among other applications, the rdf is used in the cloud microphysics community to inform process rates (i.e. as one component of turbulent collision-coalescence kernels) and diagnose mixing state. Other methods for calculating the rdf either assume an infinite domain (i.e. in direct numerical simulations of cloud drops with periodic boundary conditions) or rely on questionable assumptions to derive the rdf from effectively 1-dimensional* in situ *cloud probe observations. Given that one of the authors has been a key player in the development of a probe that can measure a truly 3d cloud volume, it makes sense that the derivation of the rdf should be revisited. That said, I am not convinced that this study will find broad application beyond the small community of digital holography observationalists.*

We are encouraged by the reviewer's apt summary of the manuscript and the fact that the reviewer feels the topic merits visiting. We do agree that the breadth of the application may be limited, though we are encouraged that there are many applications outside of the atmospheric sciences that depend on knowledge of the rdf (astrophysics,

x-ray diffraction, etc.).

*The derivation of the effective volume rdf is a rather intuitive solution. While the algorithm for computing it may be "inelegant" (authors' description), barring the development of new instrumentation with cavernous sampling volume, complex geometry and/or extremely fine pixel resolution, the brute force method of computing volume normalization factors seems sufficient.*

We agree.

*It is unfortunate that only 8 holograms were analyzed, and I would ask that the authors comment on whether there exists a threshhold concentration below which $g(r)$ cannot be accurately computed. Holograms with lower concentration will likely increase the "noise floor" length scale where $g(r)$ begins to deviate from unity but increasing the number of samples may offset this effect. If there does exist a low concentration limit, does that imply that this analysis is only suitable for a small subset of relatively high drop concentration environments?*

We refer to our introductory statement. Exactly the question raised by the reviewer is part of the other manuscript we have under review. The answer depends specifically on the instrument of choice and the properties of the sampled cloud. By way of preview, we have found that no individual HOLODEC hologram has a large enough number density of cloud droplets to reliably estimate $g(r)$ on sub-mm scales by itself; statistically reliable results will always require a number of different holographic sample volumes to be examined.

*I can envision clustering properties having some dependence on concentration, especially in light of other work using the $\Pi$ chamber (e.g. Chandrakar et al. 2016; Desai et al. 2018). Finally, it is unclear to me how application of this analysis to* in situ *measurements furthers the development of microphysical parameterizations in absence of collocated measurements of turbulence intensity or supersaturation state, and I invite the authors to expand on their vision for how this may be accomplished.*

In the other manuscript, this analysis has been completed for a field campaign where turbulence and humidity conditions were measured.

*These concerns are all relatively minor though, and I recommend this study for publication in AMT.*

We are very encouraged that this reviewer recommends this manuscript for publication.

*Specific Comments*

*Comments are given as "page X, line/figure Y" and are listed in order from beginning to end of the manuscript.*

We have addressed all of the following without exception; thank you for your help in improving our paper.

- *P.6 Fig. 1 caption: Incorporate the narrative portion of this caption into the main text. The information given here is important, and reading it from the figure caption makes it more difficult to connect it with what's happening in the text (i.e. 4th paragraph of section 3.3).*

  We have streamlined the figure caption and incorporated much of the caption's original content into the main text.

- *P. 9, Fig. 3: It's difficult to differentiate the blue and black markers unless the figure is magnified. Consider choosing more strongly contrasting colors or using different markers. Also, I assume "a.u." stands for arbitrary units but it took me a while to figure this out – please define this in the caption.*

  Good point. We have replaced figure 3 with a new version of the figure that uses a stronger color contrast. In addition, we changed a.u. to arb. units to make the $x$-axis clearer.

- *P.10, L. 18-19: Is there any study you can cite re: fragmentation near the optical windows? Is this a problem for liquid, ice, or all particles?*

The ability to detect and eliminate shattering artifacts is one of the strengths of digital holography. The approach has been thoroughly described and evaluated using airborne data (Fugal and Shaw, Atmos. Meas. Tech. 2009). Most shattering artifacts described in that paper arose in ice clouds.

The experimental data utilized in the related study alluded to previously showed a clear signature of enhanced detected droplet concentrations near the windows consistent with likely fragmentation, hence the motivation for using the sub-volume described in this study. In order to address the reviewer's request, we have added an additional reference to the Fugal & Shaw paper at this point in the manuscript.

- *P. 11, L. 9-10: I agree that a quantitative comparison with the small sample here would not be appropriate, but is it worth adding a theoretical Matèrn process to Fig. 6 for qualitative comparison with the mean curve? Disregard this comment if adding a theoretical curve distracts from the point you're trying to make.*

Upon rereading, we realize this may have been misleading. To our knowledge, nobody actually expects true *in situ* particle positions to follow a Matèrn distribution. In the previous sections the Matèrn distribution was introduced only because it is one of the systems where an analytic form for $g(r)$ is known and thus the numerical algorithm introduced here could be explicitly tested and compared to a theoretical "known". The actual curve expected for experimental data is something of a debate in the literature, and the data is far too sparse to come up with a meaningful comparison. We leave direct comparison to theory and computation as a daunting challenge for future work.

In order to reduce confusion, we have added a little more description to the properties expected beyond just "decreasing as a function of length scale".

- *P. 11, L. 16: Is the issue that there are large uncertainties in 1d* in situ *results, or rather that it's unclear from a theoretical perspective whether they are extensible to 3 dimensions given the pile of underlying assumptions?*

Both, actually. That being said, we hesitate to emphasize the second of these two points in this manuscript since the present manuscript does not really put forth the required set of underlying assumptions to meaningfully calculate an estimate of the 3d radial distribution function from a sequence of volumetrically disjoint holograms. In short, we'd be criticizing current methods based on their assumptions and arguing for the superiority of the 3d method, without establishing the assumptions inherent to the 3d method. This seems a bit unfair, so we left this discussion for another manuscript. The first of the suggested interpretations is clear and already established in the peer-reviewed literature.

*I fully agree with the first full sentence of P.12 that measuring the 3d rdf is highly desirable, but I don't think uncertainty should be the focus unless you can quantify how much it is reduced by increasing dimensionality. As I understand, the point is that your method requires fewer assumptions be made and there is greater consistency between measurement and application of the rdf.*

Good point; without establishing the fact that three-dimensions has less uncertainty we're making a true but as yet unjustified claim. We have modified the sentence in question accordingly.

- *P.12, Fig. 6 caption: Same as Fig. 1 caption – you are using the caption to communicate information that belongs in the main text.*

We have shortened the caption and augmented the main text accordingly.

- *P.1, L. 12: Is the bold-faced '?" next to Onishi et al. (2015) a missing reference?*

Yes it was. We initially missed a comma in our BibTeX file. Thanks for catching it; it should be fixed now.

- *P.2, L. 28 & 30: Extra set of parentheses surrounding reference list.*

  Thanks: citet not citep. Fixed now.

- *P.3, L. 1: Extra right parenthesis at end of sentence.*

  Actually this entire sentence is parenthetical (going back to the previous page). Hopefully when the article is typeset this won't be across multiple pages.

- *P.4, L. 6: "...in the measurement volume. $N$ is the..."' – comma instead of a period.*

  Fixed. Thanks.

- *P. 4, L. 14: "...and $N_{ex}(r_\circ)$ are the number..." – should be "is" instead of "are".*

  Thanks again. Fixed.

- *P. 11, Fig. 4 caption: "The volume simulated here corresponds from..." – should be "corresponds to"*

  Fixed.

- *P.14, L. 11: "calculate the volume of the $n$-dimensional sphere of the shell" – I'm confused by this, did you mean to say "sphere of the shell?"*

  Thanks. We meant to say spherical shell. Fixed now.

- *P.14, L. 19: Is there a factor of $1/N$ missing from the equation or is it just a straight sum?*

  It is actually just a straight sum. The notation is probably a little confusing, but equation 6 already has the $1/N$ term inside it. The quantity $g(k, j)$ isn't really a properly normalized rdf itself; it has a factor of $1/N$ "too much" divided out, so when the sum of $g(k, j)$ is taken no division by $1/N$ is needed.